# Breastfeeding at 1, 3 and 6 Months after Birth according to the Mode of Birth: A Correlation Study

**DOI:** 10.3390/ijerph17186828

**Published:** 2020-09-18

**Authors:** Irene Agea-Cano, Manuel Linares-Abad, Antonio Gregorio Ceballos-Fuentes, María José Calero-García

**Affiliations:** Department of Nursing, Faculty of Health Sciences, University of Jaén, 23071 Jaén, Spain; mlinares@ujaen.es (M.L.-A.); ancebafu@gmail.com (A.G.C.-F.); mjcalero@ujaen.es (M.J.C.-G.)

**Keywords:** breastfeeding, lactation, newborn, parturition, breastfeeding duration, mothers

## Abstract

Background: Breastfeeding is a determinant of child and maternal health. However, evidence is limited on how mode of birth influences breastfeeding. Research aim: To examine the mode of birth and breastfeeding duration and the type of lactation at one, three and six months after birth in XXX, during 2017. Methods: Correlation study on breastfeeding duration and type of lactation during the six months after birth, and mode of birth, in a randomised sample. Women ≥18 years of age with term singleton infants, were included. Collected data through interviews and hospital records. Pearson’s and Spearman’s correlation analyses were conducted. SPSSv21 and α = 0.05 were used. Results: Breastfeeding duration was shorter in women with greater parity (−0.055 **) (*p* < 0.01) and epidural analgesia (0.057 **) (*p* < 0.01), and longer in mothers with episiotomy (−0.267 **) (*p* < 0.01). Episiotomy was associated with breastfeeding at one month (0.112 **) (*p* < 0.01), and at six months (0.347 *) (*p* < 0.01). The prevalence of breastfeeding was lower in women who received epidural analgesia at three months (−0.140 **) (*p* < 0.01) and higher at six months (0.013 **) (*p* < 0.01). The percentages of breastfeeding at three months were significantly greater in women with no perineal tears (2.1) (*p* < 0.05). At six months, small rates of breastfeeding were found in women with greater parity (0.051 **) (*p* < 0.01). No significant association was detected, neither between the type of lactation and the mode of birth, nor between breastfeeding duration and the mode of birth. Conclusions: Epidural analgesia, episiotomy, perineal tears and parity influence the type of lactation and duration of breastfeeding during the six months after birth. The results suggest no association between the type of lactation and the mode of birth or between breastfeeding duration and the mode of birth.

## 1. Introduction

Breastfeeding is the essential physiological nutrition pattern for the attainment of an optimal and healthy growth and development, as stated by the International Pediatric Association (IPA), the American Pediatric Association (AAP), the Spanish Pediatric Association (AEPED), the World Health Organization (WHO), the Initiative for the Humanization of Birth and Lactation Assistance (IHAN), the United Nations International Children’s Emergency Fund (UNICEF) and the European Society for Pediatric Gastroenterology, Hepatology and Nutrition (ESPGHAN) [1]. The scientific evidence has shown a larger number of diseases, as well as longer and more serious cases of such diseases, in those children who have not been breastfed, with these effects lasting several years. Breastfeeding for longer than three months can reduce the risk of otitis media by up to 77%, the risk of lower respiratory tract infection by up to 75%, the risk of asthma by 40%, and the risk of atopic dermatitis by 42% [2]. If the breastfeeding is prolonged for over six months, it can reduce the risk of leukaemia by 20% and the risk of sudden death by 36%. Moreover, breastfeeding has been associated with a decrease in the risk of coeliac disease, obesity and type I and type II diabetes in the adult age [2]. For the mother, there is evidence of the benefits of prolonged and exclusive breastfeeding on the reduction of the risk of ovary and breast cancer [3]. Furthermore, it can help to prevent type II diabetes in women with previous gestational diabetes mellitus [4]. In addition, breastfeeding also increases the family and national resources, as it is a safe way of feeding with no environmental risks [3]. The WHO recommends initiating breastfeeding as soon as possible and maintaining it as the only way of feeding the child for the first six months after birth; then, breastfeeding should be adequately complemented with other foods, maintaining breastfeeding up to a minimum of two years after birth [3]. In Spain, there is no official monitoring system, and the data are gathered from regional surveys and the Spanish Health Surveys [5]. Most reports about breastfeeding correspond to the early neonatal period, gathered in the databases of the health centres, and they refer to hospital discharge and the type of lactation at the time of screening for metabolopathies (3–5 days after birth) [6]. Knowing the prevalence of breastfeeding, as well as its evolution over time, is essential for establishing health promotion policies, in line with numerous national strategies, such as the Strategy for Health Promotion and Prevention in the National Health System [7] and the NAOS Strategy [8] (Nutrition, Physical Activity and the Prevention of Obesity). In Andalusia, the promotion of breastfeeding, adequate feeding and other healthy habits is included in the Andalusian Primary Healthcare Service [9] and the Andalusian Integral Plan to Fight Obesity [10]. Moreover, several studies associate the duration of breastfeeding with factors such as the mode of birth, the age of the mother, parity, education about breastfeeding and epidural analgesia [11,12,13,14,15,16,17,18,19,20,21,22,23,24,25,26,27,28,29,30,31]. Our research hypothesis is that some birth variables influence the type of lactation and the duration of breastfeeding over time. The present study aims to provide results about the temporal evolution and duration of breastfeeding as a function of a set of birth variables, which has not been thoroughly studied in the described population in Spain. The aim was to analyse the relationship between mode of birth and duration of breastfeeding and the type of lactation reported by mothers at six months after birth attended to in the San Juan de la Cruz Hospital, Úbeda, Spain, during the year 2017.

## 2. Methods

### 2.1. Design

A descriptive, longitudinal and correlational study was carried out to examine the birth variables at the time of birth (age, parity, mode of birth, epidural analgesia, episiotomy, perineal tears, sex of newborn, newborn weight and whether the women had attended sessions of breastfeeding education), the type of lactation at 1, 3 and 6 months and the average duration of breastfeeding. In addition, sequential observations of the variable type of lactation at 1, 3 and 6 months after birth, in a randomised sample, were conducted.

### 2.2. Ethical Considerations

The Research Ethics Committee of Jaén resolved and ruled in favour of this study on 29 September 2016, regulated by Andalusian Decree 439/2010 of December 14th, thus complying with article 12 of Law 14/2007 of July 3rd on Biomedical Research. The study was conducted in compliance with Law 14/2007 of July 3rd of Biomedical Research, following the precepts included in the Belmont Report and the Declaration of Helsinki (updated at the Brazil Summit 2013) for biomedical research. The authors also considered Law 41/2002 on patient autonomy. The personal data of the participants were treated in compliance with Organic Law 15/1999 of December 13th on the Protection of Personal Data, informing the participants about their ARCO rights (access, rectification, correction and opposition). The anonymity of the participants was guaranteed, as well as the confidentiality of their data, which could not be accessed by people outside of the study, dissociating the personal data for their adequate protection during the data analysis.

### 2.3. Participants

The study population was constituted by women giving birth to a full-term single healthy infant. The target population was set at 1172 women who gave birth in the year 2016 in the Hospital of Úbeda (Spain). The minimum number of mothers to interview was 100 for centres with over 300 infants under 2 years of age, according to the criteria established by the IHAN [32]. Considering the number of births in the hospital, and contemplating the possible losses (20%), the sample size was set at 120 participants. To obtain a randomised and representative sample, a systematic sampling was applied on a list of births, with a sampling constant K10, by including one participant for every tenth baby born in the hospital. The inclusion criteria were the following: women aged 18 years or older, with term birth (37–42 gestation weeks) and single newborn. The exclusion criteria were women whose newborn had been admitted to the neonatal unit due to a clinical situation of the infant, maternal pathology that does not allow breastfeeding and newborns from other centres.

### 2.4. Study Measures

#### 2.4.1. Sociodemographic Variables

Some of the variables collected were mother age, parity and breastfeeding education. Parity was referred to as the number of previous deliveries, classified as primiparous (no previous births) and multiparous (at least one previous birth)). The mothers were asked whether they attended educational sessions about breastfeeding during pregnancy.

#### 2.4.2. Independent Variables

The independent variables related to the characteristics of birth were: mode of birth, epidural analgesia, episiotomy, perineal tears, sex of newborn and newborn weight.

The mode of birth was classified as vaginal birth, Cesarean birth and obstetric birth. In this paper, obstetric birth is referred to as instrumental birth. The variables epidural analgesia and episiotomy were measured by two categories (“yes” and “no”). The type of perineal tear was classified as first-degree (Type I), second-degree (Type II) and third- and fourth-degree tears (Type III), according to the Royal College of Obstetricians and Gynaecologists [33]. The sex of the newborns was categorised as male and female. The birth weight was measured in grams.

#### 2.4.3. Dependent Variables

The dependent variables were the type of lactation and duration of breastfeeding.

The type of lactation is defined by the criteria of the Spanish Health Survey [5]. Natural feeding refers to the feeding of the infant with only breast milk (known as breastfeeding); mixed lactation is the one that combines breast milk and infant formula, and formula feeding is that in which the infant is fed only with infant formula [5]. This is a categorical polychotomous variable with the following values: breastfeeding, mixed lactation and formula feeding. The duration of breastfeeding was measured by interviews and expressed in number of days.

### 2.5. Data Collection Procedure

The data were obtained through structured interviews via phone call with women who had given birth in the year 2017 in Úbeda (Spain). The birth records of the hospital provided data about the characteristics of the sample and birth variables. The women were asked about the type of lactation (breastfeeding, mixed lactation or formula feeding), describing each of them. Breastfeeding: only breast milk; mixed lactation: combination of breast milk and formula or artificial milk; and formula feeding: only formula or artificial milk (the correct term is “formula milk”, but the word “artificial” is added to avoid other interpretations). The women were not asked about the use of a bottle, since it can be used to administer any kind of milk. Repeated measurements of the study variables were recorded at three time points (1, 3 and 6 months after birth), with the aim of determining the prevalence of breastfeeding in the sample and its evolution over time. Likewise, they were asked whether they attended educational sessions about breastfeeding. The pertinent permits were requested to gain access to the database of birth records of the hospital of Úbeda (Jaén, Spain). The principal investigator valued whether the participants met the inclusion criteria and did not meet any of the exclusion criteria. Data from the year 2017 were gathered during the first six months of the year 2018, and were subsequently loaded into an anonymised database. During the months of January, February and March 2018, the authors phoned the women who had given birth between 1 January 2017 and 30 June 2017. During the months of April, May and June 2018, the authors interviewed those women who had given birth in the second half of the year 2017. The women were informed about the purpose of the study, and their written consent was requested and collected. To ensure a sample size of 120 women, potential participants were discarded after four failed phone call attempts, subsequently selecting the next candidate in the list. All women included completed the three interviews.

### 2.6. Data Analysis

A descriptive analysis was conducted for each of the variables of the database. The categorical variables are presented in a frequency table (number of cases and percentage), with the quantitative variables being expressed as mean, standard deviation, and minimum and maximum values. The description of the variable type of lactation with repeated measures on the same sample showed the evolution of such variables over time, in the first six months after birth (at 1, 3 and 6 months). A correlation analysis was carried out to determine the possible relationships between the study variables, as well as the significance and direction of such relationships. With the aim of exploring the relationships between the categorized variables and the numerical indices, the Spearman’s correlation coefficient was used, and when there were correlations between quantitative variables, the Pearson product-moment correlation coefficient (PPMCC) was used. Regarding the comparison between the mean values, the Student’s *t*-test did not show any significant results. To evaluate the differences between the type of lactation and the mode of birth, as well as between the type of lactation and the rest of the birth variables used in this study, the Chi-square test was applied, using contingency tables. Those values below 5% were considered significant. All analyses were performed using the SPSS v21 statistical software (SSPS v.21 (IBM Corp. Released 2012. IBM SPSS Statistics for Windows, Version 21.0. Armonk, NY: IBM Corp; license to use University of Jaén), Jaén, España.)

## 3. Results

### 3.1. Description of Sample and Study Variables

Descriptive data of the sample and study variables are shown in Table 1.

### 3.2. Evolution of the Type of Lactation in the First Six Months after Birth

The type of lactation at one, three and six months after birth is shown in Figure 1. Breastfeeding was the choice of 78.8%, 67.7% and 37.4% of the women at one, three and six months after birth, respectively, showing a decrease over time. Mixed lactation was the choice of 11.1%, 14.1% and 24.2% of the women at one, three and six months after birth, respectively, thus showing an increase along time. The percentage of infants who only received infant formula was 10.1%, 18.2% and 38.4% at one, three and six months, respectively, showing an increase along time. The evolution of the type of lactation in the first six months after birth is shown in Figure 2. A negative correlation was detected between the prevalence of breastfeeding and the time elapsed since the first day of feeding; thus, the longer the time, the lower the percentage of women applying breastfeeding. At six months after birth, a point was observed at which the prevalence of breastfeeding and formula feeding was similar.

### 3.3. Correlation between the Study Variables

Table 2 shows the results referred to correlations between the study variables.

#### 3.3.1. Type of Lactation at One Month after Birth

The type of lactation at one month after birth was significantly related to episiotomy (0.112 **) (*p* < 0.01) (Table 2), showing a greater value for breastfeeding. Similarly, the absence of type II perineal tear was more strongly associated with breastfeeding (corrected residuals −2.2 *p* > 0.05) (Table 3). Mixed lactation was related (2.2) (*p* < 0.05) (Table 3) to type II perineal tear. Lastly, no relationship was found between the type of lactation and the mode of birth (vaginal birth, obstetrical birth or caesarean birth).

#### 3.3.2. Type of Lactation at 3 Months after Birth

The type of lactation at three months after birth was associated with the use of epidural analgesia (−0.140 **) (*p* < 0.01), showing a lower value for breastfeeding. The percentages of breastfeeding at three months shown in Table 3 were significantly greater in women with no perineal tears (2.1) (*p* < 0.05), with a lower number of women of first-degree tears (−3.5) (*p* < 0.01), compared to the women who applied formula feeding. No relationship was found between the type of lactation at three months after birth and the mode of birth (vaginal birth, obstetric birth and caesarean birth).

#### 3.3.3. Type of Lactation at Six Months after Birth

The type of lactation at six months after birth was significantly related to some variables. The prevalence of women applying breastfeeding at six months after birth was higher in women who had been treated with episiotomy (0.347 *) (*p* < 0.05) and those who had been administered epidural analgesia (0.013 **) (*p* < 0.01). On the other hand, the prevalence of breastfeeding was lower in women with greater parity (0.051 **) (*p* < 0.01). Contingency Table 4 shows that formula feeding was associated with the presence of first-degree tears (2.9) (*p* < 0.01). The mode of birth (vaginal birth, obstetric birth and caesarean birth) was not associated with the type of lactation at six months after birth.

#### 3.3.4. Duration of Breastfeeding

The average duration of breastfeeding was 111.48 days, with an SD of 68.449 days (0–180). The duration of breastfeeding was associated with parity (−0.055 **) (*p* < 0.01) (Table 2), thus, the greater the parity, the shorter the duration of breastfeeding. Multiparous women attended fewer breastfeeding education classes (0.271 **) (*p* < 0.01) (Table 2). Although no association was found between education and breastfeeding, this condition may influence the abandonment of breastfeeding in this group of mothers. A relationship was detected between breastfeeding and episiotomy (−0.267 **) (*p* < 0.01) (Table 2). The absence of a perineal tear was significantly associated with breastfeeding at 3 months (corrected residuals: 2.1, *p* < 0.05) (Table 3). The practice of episiotomy reduced the presence of perineal tears (0.356 *) (*p* < 0.05) (Table 2). The effect of episiotomy on the duration of breastfeeding may be mediated by the absence of perineal tears. This condition can be prevented with other measures such as perineal massage, which has been proven effective in preventing tears, although not in episiotomies [34]. Studying these factors in depth would provide knowledge that may improve healthcare practice.

On the other hand, epidural analgesia was associated with a shorter total duration of breastfeeding (0.057 **) (*p* < 0.01) (Table 2). The duration of exclusively breastfeeding was associated with the type of lactation, showing a shorter duration related to the use of formula feeding in the first month (−0.602 **) (*p* < 0.01) (Table 2) and at three months after birth (−0.846 **) (*p* < 0.01) (Table 2). In the vaginal births, the average duration of breastfeeding was 112.14 days, with an SD of 68.04 (0–180). In the obstetrical births, the average duration of breastfeeding was 131.09 days, with an SD of 70.598 (0–180). In the caesarean births, the average duration of breastfeeding was 82.38 days, with an SD of 68.073 (0–180). The differences observed between the mode of birth and the duration of breastfeeding were not statistically significant, however there is an association between the use of epidural analgesia and obstetric birth (−0.279 **) (*p* < 0.01), which could explain that epidural analgesia decreases breastfeeding duration. Thus, it is not possible to assert the existence of a relationship between these two variables. Moreover, no relationship was found between the mode of birth and the type of lactation at any time point, neither at one, three nor six months after birth.

## 4. Discussion

The average age of the study population was 32.16 years, which is in line with the average maternal age in Spain (32.20 years, Spanish Statistics Institute) [35] and Andalusia (31.68 years) [35] recorded for the study year 2017. The average duration of breastfeeding was 111.48 days, with an SD of 68.449 days (0–180), which is in agreement with that reported in other studies [11,12]. The evolution of breastfeeding throughout the first six months after birth coincides with the descending trend observed at the Andalusian and Spanish level. However, the percentages of breastfeeding women in this study are higher than the average Andalusian percentages and similar to the average Spanish percentages. In our study, the percentage of women applying breastfeeding at three months after birth was 67.7%, which is above the average percentage in Andalusia (46.88%) [36], and similar to that reported at the national level by the National Health Survey for the year 2017 (63.87%) [5]. At the European level, 25% of infants received exclusively breastfeeding [37]. Our data show 37.4% for this variable, which is similar to the 39% reported at the Spanish level [5] and greater than the 20.20% reported at the Andalusian level [36]. These differences can be due to the absence of clear consensus on the measurement systems used in Spain, or to the different management of the health services of the different autonomous communities. The results of this study showed that there were differences between the mode of birth and the type of lactation during the first six months after birth. Several authors, such as Sacristan et al., did report significant differences between these two variables, stating that the infants from a vaginal birth received exclusively breastfeeding at a higher rate than those from instrumental or caesarean births (OR: 2.54, CI95%: 1.09–5.93) [12]. However, these results refer to the beginning of breastfeeding, but not to its prevalence. Other authors have separately analysed spontaneous and induced birth; in this sense, Martínez-Galiano [13] observed that, in induced birth, the risk of abandonment was 5.6 times greater than in spontaneous births (OR: 5.6, CI95%: 1.107–28.322) [13]. According to Santacruz-Salas et al. [11], in women who had a caesarean section, breastfeeding failure was 4.6 times greater than in those who had a vaginal birth (OR: 4.6, CI95%: 1.7–12.8) [11], which is in line with the results reported by other authors, such as Fernández-Cañadas [14], Brown and Jordan [15] and Carlander et al. [16]. Other authors have separately studied urgent and programmed caesarean births; with the latter being the one with greater risk of interruption of exclusively breastfeeding with respect to urgent caesarean births [17,18,19]. In the women with previous caesarean birth, authors such as Regan et al. [20] observed a lower success of breastfeeding than in the cases of birth after a programmed caesarean or urgent caesarean (OR: 1.47, CI95%: 1.35–1.60) [20]. Hobbs et al. [19] observed this difference also in comparison with vaginal birth (OR: 1.61, CI95%: 1.14–2.26) [19]. In the present study, the cases of women with caesarean birth were analysed without distinguishing between programmed and urgent caesarean, due to the small number of cases in the sample (9.1% of all births). Similarly, Rabiepoor et al. [21] reported no significant differences between the breastfeeding success rate and the type of birth. The results of other authors, such as Sharifi et al. [22], are also in line with the results of the present study, although they did not differentiate between vaginal and obstetric births. Regarding the factors related to breastfeeding success, our study found significant differences between the type of lactation during the first six months after birth, and birth variables such as epidural analgesia, episiotomy, perineal trauma and parity. The present study did not reveal any relationship between maternal age and the prevalence of breastfeeding, which is in line with the results of other studies in the Spanish population [23,24]. However, some studies have found that breastfeeding decreases with maternal age [11,25,26,27], especially after the age of 35, according to Miñano et al. [27]. Santacruz-Salas et al. [11] asserted that, in women between 36 and 40 years of age, the probability of not meeting their breastfeeding expectations was 7.5 times higher than in women under 25 years of age (OR: 7.5; CI95%: 1.8–30.9) [11]. In the present study, the absence of perineal trauma was associated with better results in breastfeeding in the first, third and sixth month after birth. Similarly, episiotomy was related to better percentages of breastfeeding at one and six months, as well as to longer total duration. The association of perineal trauma and episiotomy with breastfeeding has been poorly studied. Some authors, such as Solís et al. [28], have not found a significant association of the initiation of breastfeeding with the application of episiotomy or with perineal trauma, which is in line with our results. The results of this study show that the primiparous women had a greater prevalence of breastfeeding at six months and longer total breastfeeding duration. This is in agreement with the findings of Gil [23], who concluded that being primiparous is associated with a greater initiation of breastfeeding (*p* < 0.05), whereas being secundiparous is associated with a lower initiation (*p* < 0.01). However, other authors [18,24,29], such as Fernández-Cañadas [18], state that being primiparous is associated with the termination of breastfeeding (OR: 1.61, CI95%: 1.05–2.46) [18]. Our study showed differences between breastfeeding education and duration, although these were not statistically significant. On the other hand, such differences were statistically significant in other studies [11,13,23,29], which concluded that having no education about breastfeeding increases the risk of breastfeeding failure, as stated by Santacruz-Salas et al. [11] (OR: 9.2; CI95%: 3.0–27.9). The present study showed that the administration of epidural analgesia produced a lower percentage of breastfeeding at three months and a lower average breastfeeding duration. Our results are in line with those of authors such as Herrera-Gómez et al. [24], Lind et al. [30], and French et al. [31]. Lind et al. [30] stated that the mothers who received epidural analgesia had more probabilities of terminating breastfeeding early, regardless of the mode of birth. In this sense, French et al. [31], in a systematic review on this topic, concluded that epidural analgesia can cause breastfeeding problems in half of the women, although it is necessary to conduct a global valuation of the birth conditions. This can explain the fact that, at six months after birth, there was a significant association between epidural analgesia and greater breastfeeding (0.013 **) (*p* < 0.01) in our study.

### Limitations

This study was focused on women over 18 years of age, without pathologies, who gave term birth to a single infant, thus, the results cannot be applied to other populations. Future studies should explore, in more detail, each of the factors related to the initiation and prevalence of breastfeeding. The sample size can be a limitation to detecting a meaningful association. The mode of birth could be analysed more thoroughly, in terms of the type of initiation of the birth: spontaneous or induced, and programmed or urgent and repeated caesarean sections. The present study provides results about the temporal evolution and duration of breastfeeding, which has not been widely studied, and allowed us to estimate the prevalence at different time points in the same sample and its relationships with the birth determinants. Authors should discuss the results and how they can be interpreted in the perspective of previous studies and of the working hypotheses. The findings and their implications should be discussed in the broadest context possible. Future research directions may also be highlighted.

## 5. Conclusions

Despite the above limitations, we can reach the following conclusions:

The type of lactation during the first six months after birth is associated with birth variables such as epidural analgesia, episiotomy, perineal tears and parity.The duration of breastfeeding increases in women with episiotomy and the absence of perineal tear.Breastfeeding duration is shorter in women who received epidural analgesia and women with great parity.The mode of birth (vaginal birth, obstetric birth or caesarean birth) is related neither to the type of lactation during the first six months after birth nor to the total duration of breastfeeding.In Spain, there is no official breastfeeding temporal monitoring system that follows the international guidelines to unify the indicators and methodology recommended to obtain comparable results, thus, further research on this topic is needed.

## Figures and Tables

**Figure 1 ijerph-17-06828-f001:**
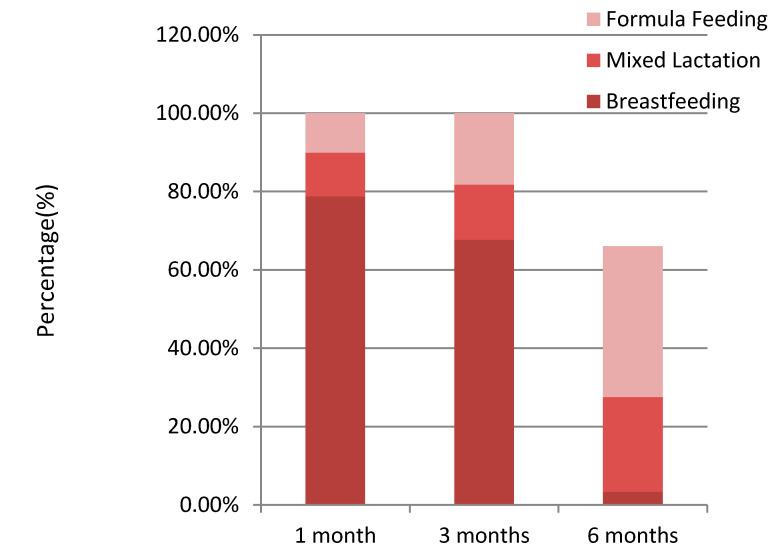
Type of Lactation at one, three and six months after birth.

**Figure 2 ijerph-17-06828-f002:**
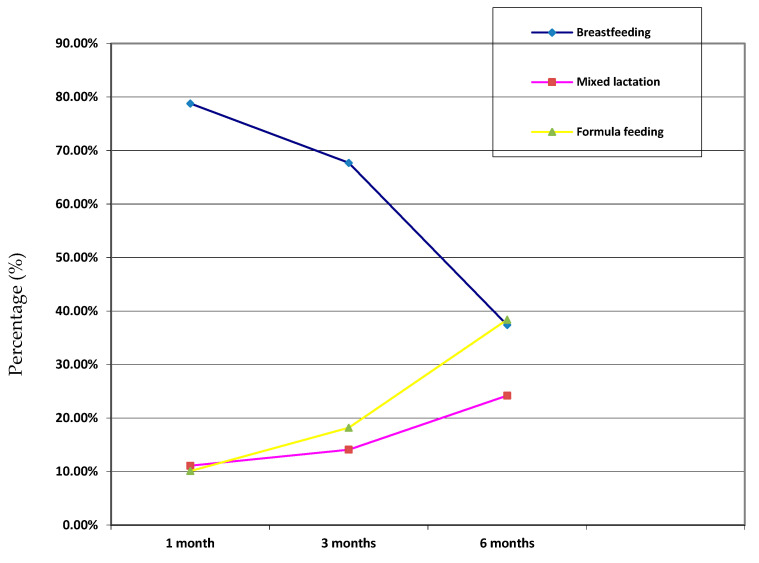
Evolution of the type of lactation in the first six months after birth.

**Table 1 ijerph-17-06828-t001:** Characteristics of the Study Population (*N* = 120).

Characteristic	*N*	%
Parity		
Primiparous	64	53.30%
Multiparous	56	46.6%
Breastfeeding Education ^a^	57	47.20%
Mode of Birth		
Vaginal	95	79.60%
Obstetric	14	11.60%
Caesarean	11	9.10%
Epidural Analgesia ^a^	57	47.40%
Episiotomy ^a^	24	20.60%
Perineal Tears		
No trauma	70	58.30%
Type I	17	14.10%
Type II	31	25.80%
Type III	2	1.60%
Sex of Newborn		
Female	72	60.10%
Male	48	39.80%

*Note: N* = 120. The average age of the participants was 32.1 years (20–42 years) (*SD* = 4.4). ^a^ Reflects the number and percentage of participants answering “yes” to this question.

**Table 2 ijerph-17-06828-t002:** Correlations between the study variables.

	MA	BE	P	EA	E	PT	MB	L1	L3	L6	BD
MA	-										
BE	-	-									
P	0.299 **	0.271 **	-								
EA	-	-	0.283 **	-							
E	0.205 *	-	-	-	-						
PT	-	-	−0.071 **	-	0.356 *	-					
MB	-	-	-	−0.314 **	-	-	-				
L1	-	-	-	-	0.112 *	-	-	-			
L3	-	-	-	−0.140 **	-	-	-	0.497 **	-		
L6	-	-	0.051 **	0.013 **	0.347 *	-	-	0.417 **	0.704 **	-	
BD	-	-	−0.055 **	0.057 **	−0.267 **	-	-	−0.602 **	−0.846 **	-	-

*Note:* MA = Maternal age; BE = Breastfeeding education; P = Parity; EA = Epidural analgesia; E = Episiotomy; PT = Perineal tear; MB = Mode of birth; L1 = Lactation at one month; L3 = Lactation at three months; L6 = Lactation at six months; BD = Breastfeeding duration. * *p* < 0.05, ** *p* < 0.01.

**Table 3 ijerph-17-06828-t003:** Contingency table: Type of lactation at one and three months with perineal tears.

		Type of Lactation 1 Month		Type of Lactation 3 Months	
	BF	ML	BF	ML	FF
Perineal Tear					
No					
Corrected residuals	1.2	−1.2	2.1	−1.6	−1.1
Type I					
Corrected residuals	1.2	−1.2	−3.5	1.0	3.5
Type II					
Corrected residuals	−2.2	2.2	0.4	1.1	−1.5
Type III					
Corrected residuals	0.4	−0.4	0.7	−0.4	−0.5

*Note:* BF = Breastfeeding; ML = Mixed Lactation; FF = Formula Feeding. Corrected typified results >1.9, *p* > 0.05. Corrected typified results >2.6, *p* < 0.01.

**Table 4 ijerph-17-06828-t004:** Contingency table: Type of Lactation at six months with Perineal Tears and Episiotomy.

		Breastfeeding	Mixed Lactation	Formula Feeding
Perineal Tears				
No	Corrected Residuals	1.0	0.4	−1.3
Type I	Corrected Residuals	−1.4	−1.6	2.9
Type II	Corrected Residuals	−0.3	1.0	−0.6
Type III	Corrected Residuals	1.3	−0.6	−0.8
Episiotomy				
Yes	Corrected Residuals	3.3	−0.6	−2.8
No	Corrected Residuals	−3.3	0.6	2.8

*Note:* Corrected typified residuals >1.9, *p* > 0.05. Corrected typified residuals >2.6, *p* < 0.01.

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
