# Peer review of "Breastfeeding at 1, 3 and 6 Months after Birth according to the Mode of Birth: A Correlation Study"

_ijerph, 2020, doi:10.3390/ijerph17186828_

Round 1

Reviewer 1 Report

Overall, the paper presents a very interesting research question that could have profound implications for public health practitioners who seek to better understand the determinants of healthy breastfeeding practices. That being said, there are grammatical and stylistic issues throughout the manuscript that make the paper difficult to follow and need to be addressed in the revisions process.

The Introduction would benefit from further explanation as to what mechanisms drive the previously identified associations between the duration of lactation and mode of birth, age of mother, parity, education, and epidural analgesia.

There are several issues that need to be addressed in the Methods:

  • The small sample size of only 120 participants seems to be too small to have the power to detect a meaningful association, so that should be addressed
  • Given that all women were recruited from the same hospital, how did you guarantee that the sample was nationally representative and that the sample did not simply reflect local/regional demographics?
  • The authors do not consider any socioeconomic or cultural factors that could influence breastfeeding access, practices, and resources.
  • The authors need to specify how all variables were measured (i.e. was birth weight in ounces?) and coded/categorized. Which categories were the reference?
  • Why did the authors choose to do a correlation study rather than regression?
  • The authors should report how many women were interviewed in each wave of the study, how many dropped out, and how many completed all three interviews. 
  • Was informed consent verbal or written?
  • Was there an association between censorship and type or duration of breastfeeding?  

Table 1 should be reformatted so that the variable is more clearly separated from its categories through bolding or indentation. This will make the table easier to read. 

Figures 1 and 2 require labels on the y-axes. 

In Table 2, all asterisks should be on the same line as the numbers. It is confusing when a second asterisk is below the number. The numbers should also be more clearly aligned with their appropriate rows.

In Table 4, it is unclear why some corrected residuals are bold while others are not.

The Results paragraphs should refer to the corresponding tables and always present the correlation coefficients. I would also recommend including the p-values within the parentheses with the coefficients.

While the Discussion does a nice job of placing this study within the literature, it does not address the reasons why these results may have been observed. Neither the Introduction nor Discussion explain the mechanisms driving the results, which is most important for readers, policymakers, and public health practitioners seeking to improve breastfeeding practices in patients.

Author Response

Response to Reviewer 1 Comments

Point 1: The Introduction would benefit from further explanation as to what mechanisms drive the previously identified associations between the duration of lactation and mode of birth, age of mother, parity, education, and epidural analgesia.

Response 1: The associations identified between the duration of lactation and mode of birth, age of mother, parity, education and epidural analgesia are explained in the Results.  We have added the mechanisms driving the obtained results.  Line number 267

Point 2:    The small sample size of only 120 participants seems to be too small to have the power to detect a meaningful association, so that should be addressed

Response 2: The sample size was calculated following the IHAN criteria, which allows comparing it with other studies. However, the sample size may be a limitation, and thus it is addressed in the Discussion. Line number 374

Point 3:    Given that all women were recruited from the same hospital, how did you guarantee that the sample was nationally representative and that the sample did not simply reflect local/regional demographics?

Response 3: In Spain, health services are public and have similar resources in obstetric services. The breastfeeding duration and mode of birth show similar rates in other hospital in Spain. However, further studies are required to compare the results.

Point 4:    The authors do not consider any socioeconomic or cultural factors that could influence breastfeeding access, practices, and resources.

Response 4: The authors considered factors that could influence breastfeeding access, such as mother age, parity and educational sessions about breastfeeding during pregnancy. In subsequent studies, the socioeconomic and cultural level will be included.

Point 5:    The authors need to specify how all variables were measured (i.e. was birth weight in ounces?) and coded/categorized. Which categories were the reference?

Response 5: The variables epidural analgesia and episiotomy were categorized by two categories (“yes”and“no”). The type of perineal tear was classified as First-degree (Type I), Second-degree (Type II) and Third-and Fourth-degree tears (Type III), according to the Royal College of Obstetricians and Gynaecologists [33]. The sex of the newborns was categorized as male and female. The birth weight was measured in grams. Line number 118

Point 6:    Why did the authors choose to do a correlation study rather than regression?

Response 6: With the results obtained in this study, we can consider regression analysis,which would allow us to develop a predictive model.

Point 7: The authors should report how many women were interviewed in each wave of the study, how many dropped out, and how many completed all three interviews.

Response 7: All women included completed the three interviews. This information has been added in the manuscript. Line number 153

Point 8: Was informed consent verbal or written?

Response 8: The women were informed about the purpose of the study and their written consent was requested and collected. Line number 151

Point 9:    Was there an association between censorship and type or duration of breastfeeding? 

Response 9: We observe no association between censorship and type or duration of breastfeeding. The study shows rates of type or duration of breastfeeding consistent with those observed in Andalusia and Spain.

Point 10: Table 1 should be reformatted so that the variable is more clearly separated from its categories through bolding or indentation. This will make the table easier to read.

Response 10: Variables are shown through bolding in Table 1. Line number 187

Point 11: Figures 1 and 2 require labels on the y-axes.

Response 11: We have added labels on the y-axes in Figures 1 and 2. Line number 202

Point 12: In Table 2, all asterisks should be on the same line as the numbers. It is confusing when a second asterisk is below the number. The numbers should also be more clearly aligned with their appropriate rows.

Response 12: There was a mistake when editing the paper. Table 2 has been improved. Line number 227

Point 13: In Table 4, it is unclear why some corrected residuals are bold while others are not.

Response 13: Table 4 has been re-edited. Line number 256

Point 14: The Results paragraphs should refer to the corresponding tables and always present the correlation coefficients. I would also recommend including the p-values within the parentheses with the coefficients.

Response 14: The correlation coefficients are presented including the p-values within the parentheses.

Point 15: While the Discussion does a nice job of placing this study within the literature, it does not address the reasons why these results may have been observed. Neither the Introduction nor Discussion explain the mechanisms driving the results, which is most important for readers, policymakers, and public health practitioners seeking to improve breastfeeding practices in patients.

Response 15: The authors address the reasons why these results were observed and explain the mechanisms driving such results, so that the readers can consider them.  Line number 267

Reviewer 2 Report

This article has the data of a very interesting story. However it is not clear. For instance the title is wrong.The breastfeeding data are  not only at 6 months also data are given on 1, and 3 months. So the title must be changed. In the abstract terms like lower and higher are used in relation to breastfeeding , that is not clear. Is the number of women lower or higher? And then in the abstract you mention perineal tears in the conclusion while nothing is said about that item before.

The introduction is ok.

In the chapter Methods: Under design, it is not clear to me what you mean.

Under the heading Participants: maybe the English can be better f.i. by women giving birth to a fullterm single healthy infant instead of gave term birth.

What do you mean with a sampling constant K 10? You choose every tenth baby born in the hospital??

Under sociodemographic variables you speak of terciparous or polyparous, it is easier to speak of primi- and multiparae.

Under results: the table 1 is fine, also figure 1. And figure 2. Maybe you can shorten the text becuase  these figures  speak for themselves.

Table 2: is confusing to me. Maybe you can do that another way?

In the discussion you give very nice results. Maybe you can do that under the heading of Results.

The English might be improved and maybe you can try to be shorter. Again you have nice data. 

Author Response

Response to Reviewer 2 Comments

REVISOR 2

Point 1: The tittle is wrong.The breastfeeding data are  not only at 6 months also data are given on 1, and 3 months. So the title must be changed.

Response 1:We have changed the title for “Breastfeeding at 1, 3 and 6 Months after Birth According to the Mode of Birth: a Correlation Study” instead of“Breastfeeding at 6 Months after Birth According to the Mode of Birth: a Correlation Study”. Line number 2.

Point 2: In the abstract terms like lower and higher are used in relation to breastfeeding , that is not clear. Is the number of women lower or higher?

Response 2: Lower and higher refer to “prevalence”; thus, higher prevalence means that there were more cases, whereas lower prevalence means that there were less cases. Line number 22.

Point 3: In the abstract you mention perineal tears in the conclusion while nothing is said about that item before.

Response 3: The Abstract shows the results of perineal tears. Line number 23.

Point 4: In the chapter Methods: Under design, it is not clear to me what you mean.

Response 4: A descriptive, longitudinal and correlational study was carried out to examine the birth variables at the time of birth (age, parity, mode of birth, epidural analgesia, episiotomy, perineal tears, sex of newborn, newborn weight and whether the women had attended sessions of breastfeeding education), the type of lactation at 1, 3 and 6 months and the average duration of breastfeeding. In addition, sequential observations of the variable type of lactation at 1, 3 and 6 months after birth, in a randomised sample, were conducted. Line number 75.

Point 5: Under the heading Participants: maybe the English can be better f.i. by women giving birth to a fullterm single healthy infant instead of gave term birth.

Response 5: Thank you for your comment, the change has been done in the manuscript: ..”women giving birth to a full-term single healthy infant”. Line number 94.

Point 6: What do you mean with a sampling constant K 10? You choose every tenth baby born in the hospital??

Response 6: We included one participant every tenth baby born in the hospital. Line number 101.

Point 7: Under sociodemographic variables you speak of terciparous or polyparous, it is easier to speak of primi- and multiparae.

Response 7: We have redefined parity as primiparous and multiparous. Line number 109.

Point 8: Under results: the table 1 is fine, also figure 1. And figure 2. Maybe you can shorten the text because these figures speak for themselves.

Response 8: We have reduced the text in the Results section as suggested. Line number 173

Point 9: Table 2: is confusing to me. Maybe you can do that another way?

Response 9: Table 2 has been improved. Line number 227.

Point 10: In the discussion you give very nice results. Maybe you can do that under the heading of Results.

Response 10: The results found have been described under the heading of Results and commented in the Discussion. The edition has been revised. Line number 267.

Point 11: The English might be improved and maybe you can try to be shorter.

Response 11: The manuscript has been revised by a professional translator.

Round 2

Reviewer 1 Report

All of the comments were addressed and appropriate edits were made.